# Uses of Inflammatory Markers for Differentiation of Intrahepatic Mass-Forming Cholangiocarcinoma from Liver Abscess: Case-Control Study

**DOI:** 10.3390/jcm9103194

**Published:** 2020-10-01

**Authors:** Sun Chul Lee, Sun Ju Kim, Min Heui Yu, Kyong Joo Lee, Yong Sung Cha

**Affiliations:** 1Department of Emergency Medicine, Yonsei University Wonju College of Medicine, Wonju 26426, Korea; bj3542@naver.com (S.C.L.); mirage0124@naver.com (S.J.K); 2SENTINEL (Severance ENdocrinology daTa scIeNcE pLatform) Team, Department of Internal Medicine, Yonsei University College of Medicine, Seoul 03722, Korea; minheuiyu@gmail.com; 3Department of Internal Medicine, Yonsei University Wonju College of Medicine, Wonju 26426, Korea

**Keywords:** laboratory diagnoses, liver abscess, cholangiocarcinoma, differential diagnoses

## Abstract

Background: Pyogenic liver abscess (LA) is difficult to distinguish from intrahepatic mass-forming cholangiocarcinoma (IMCC) in the emergency department (ED). We evaluated the predictive ability of white blood cells (WBC) and C-reactive protein (CRP) levels, neutrophil to lymphocyte ratio (NLR), platelet to lymphocyte ratio (PLR), and delta neutrophil index (DNI) in LA and IMCC in the ED. Methods: Forty patients with IMCC between January 2011 and December 2018 were included in this study. For each patient with IMCC, two control patients with LA were enrolled based on matching age and sex,—i.e., 80 patients with LA. Results: Inflammatory markers, including WBC, PLR, NLR, DNI, and CRP were significantly higher in the LA group than in the IMCC group. For both groups, the area under the curve (AUC) of the initial CRP value was significantly higher (AUC: 0.909) than that of the initial serum WBC count, PLR, and DNI levels. On multivariable logistic regression analysis with inflammatory markers, serum CRP (odds ratio, 1.290; 95% confidence interval, 1.148–1.449, *p* < 0.001) was the only significant predictor for differentiation between the LA and IMCC groups. Conclusion: Serum CRP may be a potential inflammatory marker to differentiate IMCC from LA in the ED.

## 1. Introduction

Although pyogenic liver abscess (LA) is more common in Asian countries than in Western countries [1], the incidence of LA is increasing in the United States due to the aging of the population, diabetes, hepatobiliary disease, instrumental usage of the biliary tract, and liver transplantation [2]. Despite advanced diagnostic and therapeutic techniques, the current mortality rate of LA, at 6–14%, is high [3]. The presentation is often diverse and atypical, frequently mimicking other diseases, such as intrahepatic mass-forming cholangiocarcinoma (IMCC) [4]. IMCC accounts for most intrahepatic cholangiocarcinoma [5]. Since both IMCC and LA share the features of chronic biliary inflammation and ascending cholangitis [6,7], differentiating between these two diseases can be particularly challenging in the emergency department (ED). It is important to distinguish IMCC from LA when choosing the treatment strategy, since LA is a potentially life-threatening disease [8]. Therefore, early diagnosis and timely treatment are necessary in patients with LA. Nevertheless, IMCC can mimic LA on ultrasonography (US) and computed tomography (CT) scan examination in the ED [9]. Furthermore, if IMCC is incorrectly diagnosed as LA, unnecessary treatment, such as percutaneous aspiration drainage (PAD), can result in the tumor spreading or in tumor rupture [10,11]. Therefore, it is important to identify simple, accurate, and cost-effective biomarkers for differentiating LA from IMCC that can be used in the ED to triage patients effectively.

White blood cell (WBC) levels, neutrophil to lymphocyte ratio (NLR), platelet to lymphocyte ratio (PLR), and C-reactive protein (CRP) levels are the commonly used inflammatory markers for differential diagnoses of infection in the ED. An elevated NLR has been reported to be associated with unfavorable prognosis in patients with sepsis [12], and PLR has been studied as a prognostic predictor for sepsis [13]. In addition, delta neutrophil index (DNI) was found to be a useful biomarker in the ED in various gastrointestinal diseases, such as acute pancreatitis and acute cholecystitis [14,15]. Therefore, we conducted a case-control study to determine which inflammatory markers can be used to differentiate between LA and IMCC in the ED.

## 2. Materials and Methods

### 2.1. Study Setting and Population

This retrospective and case-control observational study included consecutive patients aged > 18 years who were diagnosed with LA or IMCC and visited the ED between January 2011 and December 2018. The ED was located in a single suburban tertiary-care hospital (Wonju, Republic of Korea) that receives more than 46,000 annual patient visits and is staffed 24 h every day by board-certified emergency physicians (EPs). This study was approved by the International Review Board for Human Research (approval no. CR320047) of Wonju Severance Christian Hospital, and the study protocol conformed to the ethical guidelines of the Declaration of Helsinki (1975) and its later amendments. Since the study was retrospective and observational, informed consent was not required, and patient records and information were anonymized prior to analysis.

We initially searched for patients with a discharge code for “cholangiocarcinoma,” “liver abscess,” and “hepatic abscess,” based on the International Classification of Diseases’ tenth revision coding between January 2011 and December 2018. In addition, our radiologic database for abdominal CT examinations was searched using the search terms “cholangiocarcinoma,” “CC” (the abbreviation for cholangiocarcinoma), “mass-forming,” “liver abscess,” and “hepatic abscess,” and the search yielded 389 patients [1]. The diagnosis of LA was based on evidence of one or more fluid collections in the liver, without radiological characteristics of malignancy on US or CT studies with either one of the two following conditions: (1) identification of pus or bacteria, either microscopically or by culture, or (2) reduction in size of collection and clinical improvement in response to antibiotic administration [4]. The most common species of bacteria were *Escherichia coli* and *Klebsiella*. The major cause of LA was biliary tract infection. Treatment for LA included one of the following methods: (1) antibiotics, (2) PAD under US or CT guidance, or (3) open drainage. Initial empirical treatment usually included the administration of broad-spectrum antibiotics, with subsequent targeted therapy based on microbiological results [4]. IMCC was defined as cases confirmed by pathology when suspected lesions of cancer or mass-forming lesions in the liver were identified by a radiologist.

The study exclusion criteria included: (1) age less than 19 years (*n* = 12); (2) patients with a past history of any type of malignancy of the hepatobiliary tract (*n* = 44); (3) hematologic abnormalities, or other concurrent infections and inflammations, or treatment with granulocyte colony stimulating factors, glucocorticoids, or other immunosuppressants before study enrollment (*n* = 2) which can cause changes in inflammatory markers levels; (4) patients who refused further treatment and diagnostic procedures (*n* = 19); (5) patients transferred to another hospital after ED admission (*n* = 26); (6) insufficient data (*n* = 1) (Figure 1).

### 2.2. Data Collection

The following information was obtained from medical records: sex, age, past medical history (diabetes mellitus, hypertension, hepatitis, and previous LA), smoking, clinical symptoms, body mass index, treatments (antibiotics, PAD, chemotherapy, and surgery), and in-hospital mortality. The following factors related to the lesion were analysed on CT by a radiologist and a gastrointestinal medical expert: location, size, number, type (solid, cystic, and mixed (cyst-solid)), presence of stone, and atrophy. The following parameters were measured in the ED: inflammatory markers (WBC, neutrophil, lymphocyte, CRP (Cobas8000, Roche, Basel, Switzerland), DNI (ADVIA 2120i, Siemens Healthcare Diagnostics, Eschborn, Germany)), and levels of aspartate aminotransferase (AST), alanine aminotransferase (ALT), alkaline phosphatase (ALP), gamma-glutamyl transpeptidase (γ-GT), bilirubin, blood urea nitrogen (BUN), creatinine, carcinoembryonic antigen (CEA), carbohydrate antigen (CA) 19-9, alpha-fetoprotein (AFP), haemoglobin, and platelets. NLR and PLR were defined as the quotient of absolute neutrophil count to absolute lymphocyte count and that of absolute platelet count to absolute lymphocyte count, respectively. Particularly, DNI was acquired by the automatic calculation of the sum of immature granulocytes using parameters of ADVIA 2120i (DNI = (% neutrophil + % eosinophil) – % polymorphonuclear leukocytes) [16].

We retrospectively reviewed the patients’ electronic medical records to collect data. Data collection was conducted by an emergency physician who was blinded to the study objectives and hypothesis and trained to review charts. The abstractor was blinded to the categorization of the patient groups to reduce possible collection bias. The chart abstractor and study coordinator met periodically to review the coding rules and resolve any issues related to data collection.

### 2.3. Statistical Analysis

In all, 40 patients with IMCC were enrolled. For each patient with IMCC, we enrolled two matched simple random sampling controls with LA and no history of cancer (control group), based on age and sex. Descriptive statistics are presented as median (minimum, maximum) for continuous variables, and percentages for categorical variables. Continuous variables were analyzed by Wilcoxon ranks sum tests and categorical variables were analyzed with the chi-square test or Fisher’s exact test. To select patients, simple random sampling was performed based on age and sex. We computed the area under the curve (AUC), with 95% confidence intervals for inflammatory markers, and selected the cut-off point with Youden’s method to further compute sensitivity and specificity. A pairwise comparison of AUCs was performed using the DeLong method with Bonferroni correction. We established a multivariate logistic regression model using inflammatory markers, and odds ratios (OR) were used to distinguish the strength of diagnosis between cholangiocarcinoma and LA. Data were analyzed by R statistical software (version 4.0.1 for Windows). A *p*-value < 0.05 was considered significant.

## 3. Results

### 3.1. General Characteristics

A total of 40 patients diagnosed with IMCC and 80 patients with LA, matched by age and sex, were enrolled in this study. Table 1 shows the baseline characteristics. The participants were 58.3% male, and the median age of the study group was 63.0 years. Hypertension (33.8%) was the most common finding in the patients’ medical histories.

Fever was the most common presenting symptom in the LA group, and jaundice and weight loss were the most common symptoms in the IMCC group. In laboratory findings, ALP, γ-GT, CEA, CA 19-9, and AFP levels were significantly higher in the IMCC group than in the LA group. In addition, ALT, total bilirubin, BUN, and Cr levels were significantly higher in the LA group.

In view of the CT scan, the location of the LA was frequently the right lobe of the liver, and that of IMCC was frequently the left lobe of the liver (Table 2). There was no difference in mean size of the lesion between the two groups. The cystic lesion was frequently seen in the LA group compared to the IMCC group.

### 3.2. Comparison of Inflammatory Markers for Differential Prediction between PLA and IMCC

The levels of all evaluated serum inflammatory markers, including WBC, PLR, NLR, DNI, and CRP, were significantly higher in the LA group than in the IMCC group (Table 1). In addition, CRP showed the highest AUC level (AUC: 0.909) compared with initial serum WBC count, PLR, and DNI levels (AUC: 0.775, 0.622, and 0.796, respectively) in differentiating between the LA and IMCC groups (Table 3 and Figure 2). On multivariable logistic regression analysis, serum CRP (odds ratio, 1.290; 95% confidence interval, 1.148–1.449, *p* < 0.001) was the only significant predictor among all inflammatory markers for differentiation between the LA and IMCC groups. The cut-off level of CRP was 7.08 mg/dL, with 90% sensitivity and 90% specificity.

## 4. Discussion

To our knowledge, this is the first study to show that serum CRP levels measured in the ED can serve as a potential marker for differentiating between PLA and IMCC. In this study, among inflammatory markers, serum CRP had the highest AUC value and was the only predictor for differentiation between the LA and IMCC groups. The cut-off level of CRP was 7.08 mg/dL for predicting LA from IMCC, with 90% sensitivity and 90% specificity.

The accurate diagnosis of IMCC and LA is important to determine adequate treatments. In the ED, a CT scan is frequently performed in patients with abdominal pain, since it is a useful examination to discriminate between serious diseases and surgical abdomen. However, it is difficult to distinguish IMCC and LA by CT scan alone, since the differentiation of the two diseases can be more challenging on a single phase CT scan, such as portal venous phase, due to a lack of information regarding characteristic dynamic enhancement patterns [1]. The approach of the EPs when faced with a suspected LA is a comprehensive approach based on clinical symptoms, radiologic findings, physical examinations, and laboratory tests. In this context, it would be helpful if laboratory tests that are commonly used in the ED could help distinguish between these two clinical entities, LA and IMCC.

The typical symptoms of LA are fever, chills, and right upper quadrant pain [17]. Although most patients with LA presented with fever (82.5%), 12.5% of patients with IMCC also had fever. Similar proportions of patients in both groups presented with abdominal pain. However, the symptoms of IMCC and malignancy may be ambiguous in old age and often present with fever [18,19]. In addition, although serum CA19-9, which is identified in patients with cholangiocarcinoma [17], was significantly higher in the IMCC group, examining the tumor marker routinely in the ED is unreasonable, due to the reduced cost-effectiveness of this strategy and difficulty obtaining immediate results. Among the group of inflammatory markers, serum CRP is an acute phase reactant that increases in response to inflammatory stimuli [20]. Marked elevation of CRP levels is strongly associated with infection, including both bacterial infection and viral infection, although a stronger association exists for bacterial infection [21,22]. Nevertheless, serum CRP levels surge within 4–6 h after stimulation, double every 8 h, and peak after 35–60 h [23,24]. Although the role of serum CRP for the early detection of bacterial infection can be limited by the facts mentioned above, this weakness can be overcome, since it takes time to form an abscess. In our study, the initial CRP levels were significantly higher in the LA group compared to the IMCC group. Among the inflammatory markers, CRP showed the highest predictive value to discriminate LA from IMCC. As the level of CRP can be obtained within 30 min, it is a convenient and fast marker to use in ED. Furthermore, previous studies have demonstrated that serum CRP was useful in response-guided therapy of LA [25,26]. However, the optimum duration of antibiotic treatment is a matter of debate. Gao et al. reported that serum CRP can be considered an independent factor to determine the duration of antibiotic treatment for LA after PAD [26].

There have been several studies on the prognostic role of inflammatory markers in LA and cholangiocarcinoma. Park et al. showed that NLR is a feasible prognostic marker for LA in the ED [27]. NLR was positively associated with poor prognosis and could be a predictor of ICU admission and development of septic shock [27]. Kong et al. reported that DNI can predict the development of septic progression of LA [28]. Two meta-analyses revealed that high NLR and PLR were related to poor overall survival in patients with cholangiocarcinoma [29,30]. However, there are no studies to indicate that inflammatory markers can help distinguish between LA and IMCC. In this study, WBC, NLR, PLR, and DNI were significantly higher in the LA group compared to the IMCC group; however, only CRP proved to be an independent marker to discriminate LA from IMCC.

There have been many studies on the most common radiologic findings of LA in the right lobe of the liver, namely a well-defined, round lesion with central hypoattenuation [31]. In this study, patients with LA also had more lesions on the right side of the liver and more cystic lesions. However, radiologic findings may not be typical; rather, they may be more complex with loculated subcollections or irregular borders [7,32]. Due to these ambiguous aspects, radiologic differentiation between LA and IMCC has been a controversial and challenging topic in the literature. Indeed, the radiologic findings in this study are similar to previous reports (Appendix A). For EPs, it is extremely difficult to evaluate and determine an acute treatment plan when a liver mass is identified in the CT scan, especially when there is a need to distinguish between LA and IMCC. In an ED setting, where time is limited, inflammatory markers can provide additional information in a brief period of time.

The strength of this study includes the fact that patients were enrolled by case-control matching based on age and sex. As IMCC is a rare disease, we collected a sufficient number of patients with IMCC and LA matched by age and sex to reduce bias. In addition, this study is the first to distinguish LA from IMCC using inflammatory markers. This study also has several limitations. First, the data may have been biased since they were retrospectively obtained from a single medical center. Second, this study focused on the initial data processed in the ED, with no further discussion regarding changes in inflammatory markers. Third, although procalcitonin has recently been studied in various fields, our data were collected when procalcitonin had not yet been tested in most patients, and further study may be needed for comparison of inflammatory markers, including procalcitonin. Fourth, we could not conduct a subgroup analysis of patients with fever between the two groups, due to the small number in the IMCC group. Finally, cases in which the patient present both LA and IMCC require a careful approach on the part of the clinician to identify both entities; however, IMCC presenting primarily as liver abscess is extremely rare [33].

## 5. Conclusions

In conclusion, serum CRP is a potential inflammatory marker that can help discriminate LA from IMCC. The level of serum CRP can be obtained quickly in ED and may provide additional information to determine appropriate treatment. Further large and prospective studies are needed to confirm the predictive ability of CRP in distinguishing between LA and IMCC.

## Figures and Tables

**Figure 1 jcm-09-03194-f001:**
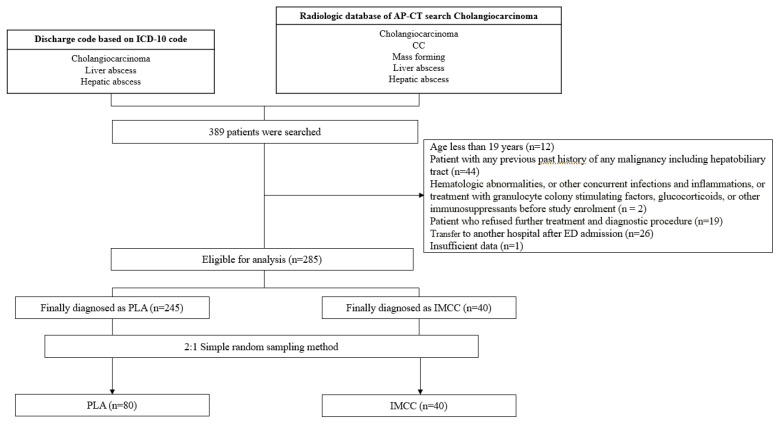
Patient flow of patients with LA and IMCC for analysis. LA: pyogenic liver abscess; IMCC: intrahepatic mass-forming cholangiocarcinoma; ICD-10: International Classification of Diseases, tenth revision; AP-CT: abdominal pelvic computed tomography.

**Figure 2 jcm-09-03194-f002:**
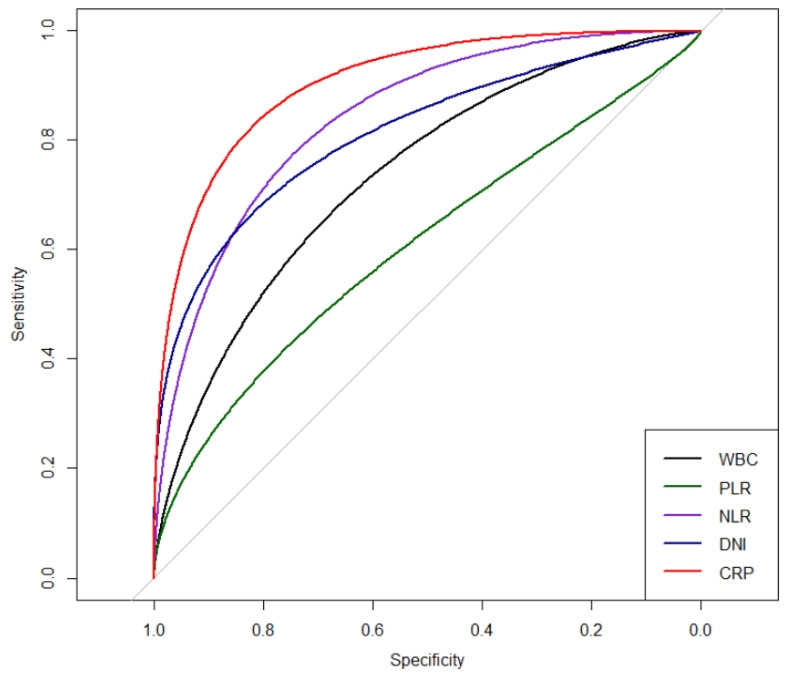
ROC curve of inflammatory markers. WBC: white blood cell; PLR: platelet–lymphocyte ratio; NLR: neutrophil–lymphocyte ratio; DNI: delta–neutrophil index; CRP: c-reactive protein.

**Table 1 jcm-09-03194-t001:** Baseline characteristics in age and sex matched patients.

Variables	Total (*n*= 120)	Liver Abscess (*n* = 80, 66.7%)	Cholangiocarcinoma (*n* = 40, 33.3%)	*p* Value
Male gender (%)	70 (58.3)	47 (58.8)	23 (57.5)	0.896
Age	63 (47–82)	64 (48–82)	62 (47–82)	0.285
Past medical history (%)				
Diabetes mellitus	30 (25.0)	18 (22.5)	12 (30.0)	0.371
Hypertension	49 (33.8)	27 (33.8)	22 (55.0)	0.026
Hepatitis	10 (8.3)	2 (2.5)	8 (20.0)	0.002*
Previous liver abscess	2 (1.7)	2 (2.5)	0 (0.0)	0.552*
Smoking (%)				0.101
Current smoker	23 (19.2)	11 (13.8)	12 (30.0)	
Ex-smoker	16 (13.3)	11 (13.8)	5 (12.5)	
Symptoms (%)				
Abdominal pain	52 (43.3)	35 (43.8)	17 (42.5)	0.86
Fever	71 (59.2)	66 (82.5)	5 (12.5)	<0.001
Jaundice	5 (4.2)	0 (0.0)	5 (12.5)	0.003*
Weight loss	4 (3.3)	0 (0.0)	4 (10.0)	0.011*
Body mass index (Kg/m^2^)	23.85(16.65–40.41)	23.88(17.62–31.39)	23.74(16.65–40.41)	0.313
Laboratory findings				
AST (U/L)	55 (31–110)	66 (34–113)	40 (28–96)	0.060
ALT (U/L)	45 (25–101)	54 (36–104)	29 (18–73)	0.001
ALP (U/L)	144 (101–222)	137 (89–214)	149 (113–309)	0.043
γ-GT (U/L)	118 (60–213)	100 (56–170)	136 (71–517)	0.007
T. bilirubin (mg/dl)	1.20 (0.24–16.04)	1.28 (0.29–4.43)	0.61 (0.24–16.04)	0.026
CEA (ng/mL)	2.00 (2.00–510.57)	2.00 (2.00–14.80)	2.66 (2.00–510.57)	0.001
CA 19-9 (U/mL)	20.40 (9.14–163.85)	10.70 (7.02–20.50)	263.60 (40.70–7197.00)	<0.001
AFP (ng/mL)	2.00 (0.30–744.42)	1.45 (0.30–5.48)	3.34 (1.40–744.42)	<0.001
Hemoglobin (g/dL)	12.45 (4.80–19.30)	12.35 (4.80–17.10)	12.75 (8.60–19.30)	0.140
Platelet (E/L)	209,500 (10,700–812,000)	176,000 (10,700–812,000)	261,000 (101,000–559,000)	0.001
Inflammatory markers				
White blood cell (E/L)	10,620 (750–24,920)	12,110 (750–24,920)	7720 (4140–18,080)	<0.001
PLR	197.55 (6.05–912.50)	218.13 (6.05–912.50)	166.53 (71.38–750.00)	0.016
NLR	8.31 (0.56–38.98)	13.02 (0.56–3.98)	3.20 (1.14–25.97)	<0.001
DNI (%)	0.65 (0.00–29.10)	2.00 (0.00–29.10)	0.00 (0.00–4.00)	<0.001
CRP (mg/dl)	13.45 (0.12–39.10)	18.95 (0.29–39.10)	1.23 (0.12–25.14)	<0.001
Biopsy (%)	41 (68.3)	1 (5.0)	40 (100.0)	<0.001
Treatments (%)				
Antibiotics	100 (83.3)	80 (100.0)	20(50.0)	<0.001
PAD	58 (48.3)	58 (72.5)	0(0.0)	<0.001
Chemotherapy	7 (5.8)	0 (0.0)	7(17.5)	<0.001*
Operation	17 (14.2)	3 (3.8)	14(35.0)	<0.001*
In-hospital mortality (%)	5 (4.2)	4 (10.0)	1(1.3)	0.024*

* Fisher’s exact test and median (min-max). AST: aspartate aminotransferase; ALT: alanine aminotransferase; ALP: alkaline phosphatase; γ-GT: gamma-glutamyl transpeptidase; T. bilirubin: total bilirubin; BUN: blood urea nitrogen; Cr: creatinine; CEA: carcinoembryonic antigen; CA 19-9: carbohydrate antigen; AFP: alpha-fetoprotein; PLR: platelet–lymphocyte ratio; NLR: neutrophil–lymphocyte ratio; DNI: delta–neutrophil index; CRP: c-reactive protein; PAD: percutaneous abscess drainage.

**Table 2 jcm-09-03194-t002:** Characteristics of computed tomography of liver abscess and cholangiocarcinoma.

Variables	Total (*n* = 120)	Liver Abscess (*n* = 80, 66.7%)	Cholangiocarcinoma (*n* = 40, 33.3%)	*p* Value
Location				
Right	78 (65.0)	58 (72.5)	20 (50.0)	0.015
Left	32 (26.7)	16 (20.0)	16 (40.0)	0.020
Both	9 (7.6)	6 (7.5)	3 (7.7)	0.970
Size	5.40 (1.50–21.00)	5.30 (1.50–14.30)	5.60 (2.30–21.00)	0.181
Number	1.0 (0.0–6.0)	1.0 (1.0–6.0)	1.0 (0.0–5.0)	0.301
Type				
Solid	32 (26.7)	3 (3.8)	29 (72.5)	<0.001
Cystic	71 (59.7)	71 (88.8)	0 (0.0)	<0.001
Cyst-solid	18 (15.1)	7 (8.8)	11 (28.2)	0.005
Stone				
GB	11 (9.2)	8 (10.0)	3 (7.5)	0.655
IHD	6 (5.0)	5 (6.3)	1 (2.5)	0.374
CBD	6 (5.0)	6 (7.5)	0 (0.0)	0.076
Atrophy				
Right	2 (1.7)	1 (1.3)	1 (2.5)	0.614
Left	11 (9.2)	5 (6.3)	6 (15.0)	0.117
Both	1 (0.8)	0 (0.0)	1 (2.5)	0.156

GB: gall bladder; IHD: intrahepatic duct; CBD: common bile duct.

**Table 3 jcm-09-03194-t003:** Comparison of inflammatory markers in discrimination of liver abscess from cholangiocarcinoma.

Markers	AUC	95% CI	Sensitivity	Specificity	Cut-Off	*p* Value
Univariate						
White blood cell	0.755	0.660–0.851	0.800	0.700	8800.00	<0.001
PLR	0.622	0.521–0.723	0.588	0.700	198.600	<0.001
NLR	0.863	0.789–0.937	0.838	0.825	5.700	<0.001
DNI	0.796	0.722–0.869	0.625	0.875	1.050	<0.001
CRP	0.909	0.850–0.969	0.900	0.900	7.080	<0.001
Pairwise comparison of AUCs						
Pairwise comparison of AUCs			difference		*p*-value*	
CRP	White blood cell		0.154		0.003	
	PLR		0.287		<0.001	
	NLR		0.046		1.000	
	DNI		0.113		0.042	
Multivariate logistic regression						
Markers	Odd ratio		95% confidence interval		*p*-value	
White blood cell /per 50	0.995		0.983–1.006		0.358	
PLR	0.994		0.987–1.002		0.134	
NLR	1.165		0.974–1.395		0.096	
DNI	1.301		0.833–2.029		0.248	
CRP	1.290		1.148–1.449		<0.001	

* Bonferroni-corrected *p*-value. AUC: area under curve; CI: confidence interval; PLR: platelet–lymphocyte ratio; NLR: neutrophil–lymphocyte ratio; DNI: delta–neutrophil index; CRP: c-reactive protein.

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
