# Peer review of "Uses of Inflammatory Markers for Differentiation of Intrahepatic Mass-Forming Cholangiocarcinoma from Liver Abscess: Case-Control Study"

_jcm, 2020, doi:10.3390/jcm9103194_

Round 1
Reviewer 1 Report
jcm-943427
The author reported the significance of CRP values in differentiating liver abscess from mass-forming type intrahepatic cholangiocarcinoma. The paper is well-written, but some points should be re-considered.
1) Generally, in the emergency department, clinical diagnosis is difficult. So useful marker for discrimination between malignancy and benign lesion. The author compared the liver abscess and cholangiocarcinoma, and some inflammatory markers show significant differences. Most important case of intrahepatic cholangiocarcinoma is not mass-forming type, but cholangiocarcinoma with ductal invasion and biliary inflammation, suggesting patient inflammatory response. Therefore, author should be compared patients with liver abscess with patients with fever or inflammatory reaction as a subclass analysis.
2) Actually, the value of CRP is important for clinical decision making.
3) The cause of liver abscess should be represented in materials and methods.
Author Response
JCM-943427
“Usefulness of inflammatory markers for differentiation of intrahepatic mass-forming cholangiocarcinoma from liver abscess: case-control study”
Point-by-point responses to Reviewer 1
We would like to thank you for the time and effort devoted to our manuscript. We have made changes in response to comments that have improved the clarity of our work.
Reviewer #1: The author reported the significance of CRP values in differentiating liver abscess from mass-forming type intrahepatic cholangiocarcinoma. The paper is well-written, but some points should be re-considered.
Major:
1.Generally, in the emergency department, clinical diagnosis is difficult. So useful marker for discrimination between malignancy and benign lesion. The author compared the liver abscess and cholangiocarcinoma, and some inflammatory markers show significant differences. Most important case of intrahepatic cholangiocarcinoma is not mass-forming type, but cholangiocarcinoma with ductal invasion and biliary inflammation, suggesting patient inflammatory response. Therefore, author should be compared patients with liver abscess with patients with fever or inflammatory reaction as a subclass analysis.
Response: Thank you for your valuable comments. As your comment, intrahepatic cholangiocarcinoma can be classified into 3 types; mass forming, periductal infiltrating, and intraductal growing. We have described it based on the following literatures (Okabayashi et al. Cancer 2001, 92, 2374-2383 and Lazaridis et al. Gastroenterology, 2005, 128(6), 1655-1667).
As your comment, ductal invasion and biliary inflammation develop inflammatory response. Therefore, we conducted a subgroup analysis of only patients with fever in both groups. However, there were only 5 patients with fever in the CCC group, so statistical analysis was limited. We have added this point to limitation section. And the statistical analysis results are attached below. In the case of DNI, there were few patients (5 patients) in the CCC group and there were many patients with 0, so result was not available in multivariate logistic regression. When we plan the next study, we will take your valuable comments into consideration.
Inflammatory markers |
Liver abscess (N=66) |
Cholangiocarcinoma (N=5) |
P value |
|||||||
White blood cell |
12180 (750-24920) |
14480 (6780-18080) |
0.839 |
|||||||
PLR |
211.45 (6.0-912.5) |
92.2 (71.4-219.7) |
0.067 |
|||||||
NLR |
12.6 (0.6-39.0) |
3.2 (2.5-11.7) |
0.036 |
|||||||
DNI |
2.05 (0-20.5) |
0 (0.0-0.0) |
0.005 |
|||||||
CRP |
19.15 (0.29-39.1) |
6.87 (1.1-25.14) |
0.055 |
|||||||
Markers |
AUC |
95% CI |
Sensitivity |
Specificity |
Cut-off |
P value |
|
|||
Univariate |
|
|
|
|
|
|
|
|||
White blood cell |
0.471 |
0.144-0.798 |
0.818 |
0.400 |
8460.0 |
0.589 |
|
|||
PLR |
0.748 |
0.574-0.923 |
0.712 |
0.800 |
154.60 |
0.034 |
|
|||
NLR |
0.785 |
0.577-0.993 |
0.939 |
0.600 |
3.250 |
0.018 |
|
|||
DNI |
0.871 |
0.818-0.924 |
0.742 |
1.00 |
0.050 |
0.003 |
|
|||
CRP |
0.761 |
0.506-1.00 |
0.909 |
0.600 |
7.010 |
0.027 |
|
|||
Pairwise comparison of AUCs |
|
|||||||||
Pairwise comparison of AUCs |
difference |
p-value* |
|
|||||||
CRP |
White blood cell |
0.290 |
0.042 |
|
||||||
PLR |
0.013 |
0.949 |
|
|||||||
NLR |
0.024 |
0.888 |
|
|||||||
DNI |
0.110 |
0.402 |
|
|||||||
Multivariate logistic regression |
|
|||||||||
Markers |
Odd ratio |
95% confidence interval |
p-value |
|
||||||
White blood cell /per 50 |
1.026 |
0.992-1.061 |
0.134 |
|
||||||
PLR |
0.988 |
0.967-1.009 |
0.257 |
|
||||||
NLR |
0.723 |
0.429-1.22 |
0.225 |
|
||||||
DNI |
- |
- |
- |
|
||||||
CRP |
0.866 |
0.703-1.068 |
0.179 |
|
||||||
- Actually, the value of CRP is important for clinical decision making.
Response: We agree with reviewer’s comment. The value of CRP is easily obtained and important to make a decision in clinical field. We already described cut-off value of CRP (7.08 mg/dL) in the manuscript.
3.The cause of liver abscess should be represented in materials and methods.
Response: Thank you for your comment. The major cause of liver abscess was biliary tract infection same as other literature (Rahimian et al. Clinical Infectious Diseases, 2004, 39(11), 1654-1659). When bacteria were detected in the LA group, the types of bacteria were Escherichia coli, Klebsiella, Streptococcus, Staphylococcus, and anaerobic organisms but are generally polymicrobial. We described the cause of liver abscess in materials and methods.
We would like to thank for your very helpful comments; we feel these have significantly improved the quality of the manuscript.
Reviewer 2 Report
In their manuscript entitled « Usefulness of inflammatory markers for differentiation of intrahepatic mass-forming cholangiocarcinoma from liver abscess: case-control study », Lee et al., investigate the use of inflammatory markers as a tool to distinguish pyogenic liver abscess (PLA) from intrahepatic mass-forming cholangiocarcinoma patients. This study is a simple case-control study and shows the usefulness of CRP levels for the identification of PLA patients that leads to rapid and more accurate diagnosis.
I have only one small comment for the authors :
- The authors discussed that detection of CRP levels is preferable over the CA19-9 for the rapid differentiation of the two groups of patients. What about the use of novel circulating non-coding RNA biomarkers in cholangiocarcinoma, which are frequently increased in cholangiocarcinoma patients (e.g. miR-21) ? miRNAs can be easily detected and amplified. Can they be used in combination to CRP for more presice identification of patients ?
Author Response
JCM-943427
“Usefulness of inflammatory markers for differentiation of intrahepatic mass-forming cholangiocarcinoma from liver abscess: case-control study”
Point-by-point responses to Reviewer 2
First of all, we would like to thank Reviewer 1 for his/her comments, which helped us to improve this manuscript.
In their manuscript entitled « Usefulness of inflammatory markers for differentiation of intrahepatic mass-forming cholangiocarcinoma from liver abscess: case-control study », Lee et al., investigate the use of inflammatory markers as a tool to distinguish pyogenic liver abscess (PLA) from intrahepatic mass-forming cholangiocarcinoma patients. This study is a simple case-control study and shows the usefulness of CRP levels for the identification of PLA patients that leads to rapid and more accurate diagnosis.
Major points:
I have only one small comment for the authors :
The authors discussed that detection of CRP levels is preferable over the CA19-9 for the rapid differentiation of the two groups of patients. What about the use of novel circulating non-coding RNA biomarkers in cholangiocarcinoma, which are frequently increased in cholangiocarcinoma patients (e.g. miR-21) ? miRNAs can be easily detected and amplified. Can they be used in combination to CRP for more presice identification of patients ?
Response: Thank you for your comments. As commented, microRNA (miR)-21 and miR-221 are potential diagnostic markers for primary intrahepatic cholangiocarcinoma (Correa-Gallego et al. PLOS ONE, 2016, 11(9), e0163699). MiR-21 expression levels accurately differentiate patients with intrahepatic cholangiocarcinoma from controls and could serve as adjunct in diagnosis (Correa-Gallego et al. PLOS ONE, 2016, 11(9), e0163699). However, the identification of miR expression is still challenging in clinical field, especially in emergency department. Though, I agree that combination of miR expression and CRP would be more precise to identify cholangiocarinoma or liver abscess.
We would like to thank for yor very helpful comments; we feel these have significantly improved the quality of the manuscript.
Round 2
Reviewer 1 Report
No further revision is need.